# Cell Culture Models for Hepatitis B and D Viruses Infection: Old Challenges, New Developments and Future Strategies

**DOI:** 10.3390/v16050716

**Published:** 2024-04-30

**Authors:** Arnaud Carpentier

**Affiliations:** 1Institute for Experimental Virology, TWINCORE Centre for Experimental and Clinical Infection Research, a Joint Venture between Hannover Medical School (MHH) and Helmholtz Centre for Infection Research (HZI), Feodor-Lynen-Strasse 7, 30625 Hannover, Germany; arnaud.carpentier@twincore.de; 2Cluster of Excellence RESIST (EXC 2155), Hannover Medical School, Hannover, Germany; 3Hannover Medical School, Carl-Neuberg-Straße 1, 30625 Hannover, Germany

**Keywords:** Hepatitis B Virus, Hepatitis D Virus, primary hepatocytes, hepatoma cell lines, HLCs, cell culture, model

## Abstract

Chronic Hepatitis B and D Virus (HBV and HDV) co-infection is responsible for the most severe form of viral Hepatitis, the Hepatitis Delta. Despite an efficient vaccine against HBV, the HBV/HDV infection remains a global health burden. Notably, no efficient curative treatment exists against any of these viruses. While physiologically distinct, HBV and HDV life cycles are closely linked. HDV is a deficient virus that relies on HBV to fulfil is viral cycle. As a result, the cellular response to HDV also influences HBV replication. In vitro studying of HBV and HDV infection and co-infection rely on various cell culture models that differ greatly in terms of biological relevance and amenability to classical virology experiments. Here, we review the various cell culture models available to scientists to decipher HBV and HDV virology and host–pathogen interactions. We discuss their relevance and how they may help address the remaining questions, with one objective in mind: the development of new therapeutic approaches allowing viral clearance in patients.

## 1. Introduction

Hepatitis viruses, named A to E, share the same host: adult hepatocytes. They, however, differ vastly in terms of biology and are clinically different from each other. From viruses causing acute self-limiting infection like HAV and HEV, to viruses leading to chronic infection and hepatocellular carcinoma (HCC) development like HCV, the field of Hepatitis virology is diverse and fascinating. One of the most intriguing observations in Hepatitis virology is the interplay between the Hepatitis B Virus (HBV) and the Hepatitis D Virus (HDV). The virology and life cycles of these two viruses (Figure 1) have been reviewed elsewhere [1,2,3,4]. Briefly, HBV is an enveloped DNA virus belonging to the Hepadnavirus family. Its genome is a partially double stranded relaxed circular DNA (rcDNA), enclosed within a viral capsid formed of the HB core antigen (HBcAg), and surrounded by a viral envelope carrying three isoforms of the Hepatitis B surface glycoprotein (named large, medium, and small HBsAg). HBV first interacts with heparan sulfate proteoglycans (HSPGs) on the surface of the hepatocyte. Then, the large isoform of the HBsAg (L-HBsAg) binds to the sodium taurocholate co-transporting polypeptide (NTCP), leading to endocytosis, membrane fusion, and release of the viral DNA into the host cell’s nucleus. There, the viral genome is repaired, forming the so-called covalently closed circular DNA (cccDNA). This intranuclear mini-chromosome-like structure then acts as a template for the cellular RNA polymerase II to allow transcription of different viral RNAs. Pre-genomic and messenger viral RNAs are brought to the cytoplasm where they are translated. The newly synthesized viral proteins participate in the assembly of new progeny viruses. Capsid proteins assemble around the viral pre genomic RNA associated with one copy of the viral polymerase. The assembled capsid then associates with the three isoforms of the HBsAg found in the endoplasmic reticulum (ER) membrane and is then released through the secretory pathway. The new viral DNA genome is then created inside the newly formed capsid by the enclosed viral polymerase. To be noted, the viral DNA can also integrate back into the cellular genome, then producing HBsAg even in absence of active viral replication.

HDV is the smallest RNA virus to infect humans. HDV is a negative sense single-stranded circular RNA virus whose genome is associated with Hepatitis Delta antigens (HDAg) to form a ribonucleoprotein (RNP). After viral entry, the RNP is brought to the nucleus, where the genome replication takes place. Its genome encodes no viral RNA-dependent RNA polymerase (RdRp). In absence of an RdRp, HDV hijacks the cellular DNA-dependent RNA polymerase type II to replicate its genome through formation of a positive strand viral “antigenome”. Importantly, this antigenome can be modified by the cellular ADAR1 enzyme. For this reason, two different mRNAs can be transcribed from the viral genome, allowing further translation of two different isoforms called small and large (S- and L-HDAg), respectively, important for genome replication and viral assembly. The HDAg is the only viral encoded protein. HDV is a deficient virus; HDV does not encode a surface antigen. To perform the later steps of viral assembly, it relies on a helper virus. HDV uses the envelope of HBV, through interaction of the L-HDAg with the ER-embedded HBsAg, allowing budding of the new viral particles surrounded by an HBV envelope. For this reason, HDV and HBV share the same receptor at the hepatocyte surface, NTCP. HDV can only fully complete its life cycle in hepatocytes co-infected with HBV or carrying integrated HBV DNA. On the other hand, upon HDV infection of naïve cells, HDV genome replication, HDAg translation and modification can be reproduced independently from HBV.

HBV causes chronic infection in 15–40% of the exposed patients, leading to cirrhosis and HCC in about 25% of chronically impacted patients, with a strong influence of the age of the patient. Five percent of chronically infected HBV patients are believed to be HDV-positive, with high variability from region to region [5]. Importantly, these numbers are probably underestimated due to lack of testing and diagnosing in many parts of the world. Co- or superinfection of HBV-infected patients with HDV leads to a more severe form of viral Hepatitis, Delta Hepatitis, characterized by accelerated disease progression and significantly higher risk of developing HCC [6]. Despite the development of prophylactic vaccines against HBV, there is still a need for efficient treatments against HBV and HDV infection [7]. PEGylated interferon alpha (IFNα) treatment shows limited efficacy. Nucleos(t)ide analogs can significantly decrease HBV replication in patients, but stopping treatment leads to rebound of the disease due to the maintenance of the HBV cccDNA in infected hepatocytes. For Delta Hepatitis patients, recent development in targeting virus–host interactions significantly improved therapeutic options. Among them, Lonafarnib inhibits the farnesylation of the L-HDAg, thus preventing its interaction with the HBsAg. More advanced is Myrcludex, now licensed in Europe under the name Hepcludex. Myrcludex targets and prevents interaction between the viral receptor NTCP and the L-HBsAg, thus blocking viral entry and stopping viral spread. Importantly, while these new treatments limit disease progression in many patients, no treatment allows clearance of HBV or HDV. 

In this context, it is important to develop relevant cell culture models to investigate HBV and HDV biology and host–pathogen interactions, and study the interplay between the two viruses in an environment as close as possible to the natural host cell, the adult hepatocyte. The field is evolving rapidly, and progress has been made in the last few years to allow better understanding of cellular mechanisms supporting HBV and HDV infection, replication, and persistence in the hepatocyte.

## 2. In Vitro Models to Study Virus Infection

Through the years, from the discovery of the viruses to the clinical approval of Hepcludex, research on HBV and HDV has been supported by the development of cell culture models that allow understanding of the virus’ host–pathogen interactions (Figure 2).

### 2.1. Primary Human Hepatocytes (PHHs)

Isolated from healthy parts of pieces of liver resection, human adult hepatocytes in primary culture (PHHs) represent the Gold Standard for in vitro hepatocyte research. They have been extensively used for toxicity and drug metabolism assays due to the maintenance of their hepatic functions, particularly their expression of detoxifying enzymes and isoforms of the cytochrome P450 [8]. 

Through the years, they have been described to be permissive to all known primary Hepatitis viruses, including HBV and HDV [9]. Interestingly, using high titer inoculum, virtually all PHHs can be infected with HBV in vitro [10], confirming they can, under certain circumstances, be an efficient model for deciphering viral entry and replication. For example, works on PHHs have demonstrated that HBV infection is facilitated by poly ethylene glycol (PEG) [11], an experimental setting nowadays used for every cell culture model. In vitro culture of primary hepatocytes (PHs) reproduces viral species tropism where freshly isolated PHs from not only human, but also chimpanzees were shown to be permissive to HBV and HDV infection in vitro [12]. In addition, HBV also infects PHs from the Northern Treeshrew Tupaia belangeri. In this context, Yan H et al. used Tupaia PHs to identify NTCP as the receptor for HBV and HDV [13], before confirming their finding in human cells. Importantly, PHHs in vitro maintain most of their hepatic functions and remain immune competent, making them an important tool to study or validate host–pathogen interactions associated with response to the infection. 

Despite all these advantages, PHHs exhibit strong limitations. Coming from clinical material, they have a very scarce availability. Thanks to the development of cryopreservation protocols, they are now commercially available, but at high cost. Their handling in laboratory settings remains difficult; they undergo fast dedifferentiation upon in vitro plating, including depolarization and gradual loss of their metabolic functions, allowing only a limited time window for experimental work. In vitro dedifferentiation of PHHs has been a major hurdle for establishing long-term and efficient in vitro replication of HBV. The hepatic transcription factor HNF4A, a major driver of the hepatic phenotype, is critical for HBV replication [14]. However, PHHs rapidly lose expression of HNF4A upon plating, making HBV in vitro infection inefficient and limited in time. Advances in culture conditions, for example by co-culturing PHHs with non-parenchymal cells, slows dedifferentiation and thus can improve efficiency of infection [15,16]. Finally, PHHs exhibit strong donor-to-donor variability, for example, related to viral susceptibility.

Despite their limitations, PHHs remain a tool of choice and the most reliable cell culture model for validation of results obtained in other experimental models.

### 2.2. Hepatoma-Derived Cell Lines Huh7/HepG2

Because the use of PHHs is limited, work on transformed hepatic cell lines has been critical for studying hepatocyte physiology and viral infection in vitro. Transformed cell lines, derived from hepatocellular carcinoma show unlimited availability, low cost of maintenance, and their amenability to most laboratory techniques makes them a tool of choice, particularly for advanced mechanistic research and high throughput works. In the field of Hepatitis virology, the two main used cell lines are Huh7 [17] and HepG2 [18], both isolated from male hepatoma tissue. Importantly, Huh7 differs metabolically from PHH. Loss of most of their mature hepatic metabolic functions, from polarization and detoxification to lipid metabolism limits the use of Huh7 for specific applications. HepG2 reproduces some aspects of the hepatic metabolism, making it a more relevant tool to study xenobiotic metabolism and detoxification. HepG2 also represents a more relevant model than Huh7 in terms of cellular polarization.

Huh7 has been shown to be a critical model to study HCV infection and its host–pathogen interactions, as they naturally support viral entry, genome replication, and productive viral infection using selected HCV isolates (Reviewed in [19]). Interestingly, Huh7 and HepG2 are not permissive to HBV and HDV in vitro infection. However, their transfection with recombinant HBV genome [20,21,22] or pgRNA [23,24] allows new progeny virus production [20,21,25]. Similarly, while these cells are not susceptible to HDV, their transfection with three head-to-tail HDV cDNA allows viral genome replication, expression of the HDAg, and production of the viral RNP [26]. Allowing production of the HBsAg in the same cells allows production of infectious HDV [27]. Altogether, it suggests that these hepatoma cell lines are deficient for cellular processes associated with viral entry of HBV, but otherwise support later stages of the viral life cycle. 

Stable transfection of the HBV genome in HepG2, for example, the cell lines HepAD38 [22] and HepG2.2.15 [21], is now commonly used for production of infectious HBV. Recently, HepG2 have been stably transfected with various HBV genotypes, allowing production of infectious HBV of Genotype A to H [28]. Similarly, lentivirus-based transduction of the sequence allowing production of the three isoforms of the HBsAg followed by transfection of the pSVLD3 containing the DNA sequence of HDV of genotype 1 allows stable production of high titer of progeny HDV (Huh7-2C8D; [29]).

Despite their undifferentiated state and inherent limitations, these cell lines remain a unique tool for cost-efficient, high throughput, or screening assays.

### 2.3. HepaRG, dHepaRG

Contrary to the aforementioned hepatoma cell lines, HepaRG was the first cell line described as permissive to HBV and HDV infection in vitro [30,31]. HepaRG is an immortalized hepatic cell line derived from an HCV-induced liver cancer. Importantly, it expresses hepatic genes and transcription factors at a more relevant level than other liver cancer cell lines [30]. For this reason, HepaRG has been widely used for drug metabolism and detoxification studies due to their maintained expression of isoforms of the cytochromes P450 (CYPs) and metabolizing enzymes [32]. Importantly, HepaRG is a liver progenitor cell line [30]: To reproduce a more mature hepatic phenotype in vitro, the cells need to be differentiated in vitro using long culture in the presence of hydrocortisone and dimethyl sulfoxide (DMSO). However, this process leads to a heterogeneous population, constituted of hepatocyte-like and cholangiocyte-like cells. Early studies have shown that HBV infection is limited to the cell population resembling hepatocytes while cholangiocyte like cells are not permissive to infection [31]. Because differentiated HepaRG (dHepaRG) showed a higher susceptibility to HBV or HDV infection than non-differentiated HepaRG, Ni et al. studied transcriptomic changes during HepaRG in vitro differentiation, and independently identified NTCP as the critical host factor and viral receptor conferring susceptibility to HDV/HBV infection during HepaRG differentiation [33]. In order to circumvent the long, heterogeneous, and sometimes irreproducible process of maturation necessary to express sufficient levels of the viral receptor, NTCP has then been over-expressed in non-differentiated HepaRG (HepaRG-NTCP) [33]. This renders the cells readily permissive to HBV and HDV.

HepaRG, however, exhibits a low infection efficiency, with between 5 and 25% of infected cells, and requires more than a week to establish robust HBV replication. Moreover, it seems that this cell population does not support HBV viral spreading. Hantz et al. pointed this limitation to a defective or slow HBV DNA repair mechanism leading to a limited establishment of the cccDNA [31]. HepaRG, however, supports other key physiological features of HBV infection, like viral integration [34]. Importantly, dHepaRG and HepaRG-NTCP exhibit a mature innate immunity [35] and respond to infection with HBV [36] and HDV [37] similarly to PHHs. It confirms them as particularly useful models to study many aspects of host–pathogen interactions.

Recent developments to obtain faster differentiation, using a chemical mix in addition to DMSO, did not improve viral susceptibility and replication [38]. Moreover, the in vitro heterogeneous differentiation process makes them inadequate for high throughput assays. Directly overexpressing NTCP in undifferentiated HepaRG partially lift this restriction, but how their lower level of hepatic maturation influences HBV and HDV replication in the cells remains to be investigated. 

### 2.4. NTCP Overexpressing Hepatoma Cell Lines

Following the identification of NTCP as the receptor for HBV and HDV [13,33], NTCP has been exogenously overexpressed in the non-susceptible Huh7 and HepG2 hepatoma cell lines (see Section 2.2), rendering them permissive to HDV and HBV entry and infection [13]. HBV infection of these cell lines allows production of progeny viruses and spreading of the infection, confirming them as particularly permissive models to reproduce the entire life cycle of HBV in vitro. However, new hepatocellular carcinoma-derived cell lines have been recently designed, supporting higher replication of HBV, suggesting that HepG2-NTCP and Huh7-NTCP hepatoma cell lines may still inherently express mechanisms of restriction limiting HBV infection [39].

Interestingly, using the same HBV multiplicity of infection, HepG2-NTCP exhibits a higher infection rate than Huh7-NTCP (~70% vs. max 5%). Intriguingly, Huh7-NTCP seems more permissive to HDV than to HBV [33]. While HDV can readily infect Huh7-NTCP, HBV necessitates treating the cells with DMSO or heparin [40] to allow more efficient infection, even when the viral receptor is highly expressed. It suggests that different hepatoma cell lines may express different levels of unknown host factors necessary to support infection and replication of HBV or HDV. It makes it difficult to establish an efficient co-infection system in these cells allowing the reproduction of the HDV whole life cycle. In this context, researchers have established hepatoma cell lines stably expressing the HBsAg to allow supporting HDV infection. The HepNB2.7 cell line is an HepG2 cell line stably expressing both the viral receptor NTCP and the HBsAg, thus allowing HDV replication and spread upon in vitro inoculation [41]. Integration in Huh7 cells of the DNA sequences of the HBV envelope proteins, of the NTCP gene, and of the HDV genome in the so-called Huh7-END, allows stable production and spread of HDV even in absence of inoculation [42].

Due to their high amenability to in vitro works, such cell culture models are extremely valuable for high-throughput assays. For example, using a highly susceptible clone of the NTCP-Huh7 cell line called Huh-106, Verrier et al. performed siRNA [43] and small molecule screens [44], leading to the identification of several new host factors of HDV and HBV.

Overall, infection of the NTCP-overexpressing hepatoma cell line is relatively efficient but requires high viral inoculum. Moreover, they lack several cellular metabolic pathways, for example related to the IFN innate immunity, and their transformed and dedifferentiated nature limits their biological relevance, making them a rather poor model to study specific host–pathogen interactions. Thus, works on hepatoma cell lines often require validation in more mature models like PHHs or dHepaRG.

### 2.5. Human Pluripotent Stem Cell-Derived Hepatocyte like Cells (HLCs)

In the last 15 years, stem cell-derived hepatocytes, called induced hepatocytes or hepatocyte-like-cells (HLCs) have been proposed as a valuable alternative to PHHs. HLCs are differentiated in vitro from human pluripotent stem cells (hPSCs), for example, embryonic or induced pluripotent stem cells [45] and hepatic progenitor cells, using various protocols based on sequential treatments with small drugs and/or growth factors [46,47]. After 15 to 25 days of differentiation, depending on protocols, HLCs display phenotype and cell functions close to PHHs. While they retain fetal and immature hepatic characteristics, like AFP expression and lower CYP activity, they constitute an interesting alternative to PHHs with increased availability and lower cost. Moreover, they are not subjected to the dramatic dedifferentiation of PHHs in vitro and are amenable to genetic modification at the pluripotent stage (reviewed in [48]).

Importantly, HLCs have been described to be permissive to all primary Hepatitis viruses (reviewed in [49]), including HBV [50,51,52] and HDV [53]. hPSCs themselves are not susceptible to HDV or HBV infection. However, during the process of hepatic differentiation, cells gradually express critical viral host factors, like the HNF4α and the viral receptor NTCP, conferring them permissivity to HBV and HDV. Importantly, mature HLCs constitutively express a high level of NTCP [53]. Thus, HLCs are naturally susceptible to HBV in vitro infection [50,51,54]. Moreover, HLCs support establishment of cccDNA and production of progeny viruses [51]. Using very high MOI, virtually all HLCs could be infected [51], similarly to PHHs [10]. Mature HLCs are also susceptible to HDV infection and support viral genome replication and transcription of the HDAg [53]. However, the model is limited by very low infection efficiency, despite expression of NTCP and other known host factors. Importantly, overexpression of HBsAg in these cells or coinfection with HBV allows production of infectious progeny HDV [55].

As aforementioned, susceptibility to HBV and HDV follows induction of NTCP expression during the hepatic specification stage [53]. In addition, Chi et al. compared HDV susceptibility at different days of hepatic maturation. They suggest that CD63, a gene differentially expressed during hepatic maturation, plays a role in promoting the infection of HLCs by HDV [55].

Importantly, HLCs can be maintained in culture for several weeks [51,56] without significant loss of expression of host factors such as HNF4A or NTCP. While maintaining HBV replication for up to four weeks, HLCs also supported viral spreading [51], suggesting they may constitute a unique mature hepatic model for HBV study. Intriguingly, HDV replication could not be maintained for more than 9 days but was partially rescued by treating the cells with the farnesyl transferase inhibitor, Lonafarnib. It suggests that the accumulation of farnesylated L-HDAg participated in the inhibition of HDV genome replication in HLCs. Future studies are needed to understand whether efficient co infection with HBV and HDV may partially rescue these limitations, by allowing excretion of L-HDAg in progeny viral particles. Moreover, establishment of a spreading HDV infection could help maintain the ongoing infection.

One critical characteristic of HLCs is that they are innate immune competent. Infection with hepatotropic viruses, like HCV, triggers a strong innate immune response similar to the one of PHHs [56], capable of restricting viral replication. Interestingly HBV infection does not [57], confirming its stealth character (reviewed in [58]), On the other hand, HDV infection of HLCs triggered an IFN- and NFkB-dependent innate immune response [53], but this response did not inhibit HDV genome replication.

In addition to 2D monoculture of HLCs, hPSCs can be differentiated with mesenchymal and endothelial cells to form 3D structure liver organoids (LOs) [59]. Hepatocytes in LOs are more mature than in monoculture and their permissiveness to viral infection has been investigated. Despite comparable expression of NTCP, LOs support better HBV replication than HLCs in 2D monoculture [60]. Importantly, the authors observed variable levels of infection depending on the genetic background of the cells used for differentiation. It opens perspectives of research for studying the effect of the genetic background of the patient on viral natural history, by using patient-derived iPSC or liver bipotent progenitor cells [61] to generate mature HBV and HDV permissive LOs [62,63].

## 3. Biological Relevance and Specific Questions to Be Answered

We saw that different cell culture models have different susceptibility to HBV and HDV infection, and do not allow the reproduction of every stage of the virus’s cycles (Table 1). Combinations of benefits and disadvantages make none of them a perfect universal tool to decipher every aspect of the virus’s life cycle and host–pathogen interactions. Moreover, these different models differ in many points (Table 2) and, depending on the question investigated, the cell culture models should be carefully considered. Some specific points and questions are discussed here.

### 3.1. Hepatic Polarity In Vitro: Effect on Expression and Access to NTCP

Hepatocytes in situ are characterized by their morphological and functional polarization [64]. NTCP, the receptor for HBV and HDV, is also associated with cell polarity. NTCP is expressed exclusively in the liver, where it mediates the uptake of bile acids from the blood to the hepatocytes. NTCP is expressed on the basolateral pole of in situ hepatocytes, facing the endothelium–blood interface. For this reason, NTCP, hepatocyte function, and cell polarity are strongly linked to each other, but how in vitro cell polarity affects HBV and HDV infection is not fully understood. Reproducing the complete 3D organization of in situ hepatocytes is impossible in vitro. Digestion of hepatic tissue disrupts cell polarity, and in vitro seeding of PHHs in a monolayer only partially reproduces hepatic polarization, with only part of the cells maintaining functional apical pole and formation of bile canaliculi [64]. Importantly, cell transformation (as in Huh7 and HepG2) is associated with loss of cellular and functional polarity. While HepG2 maintains some aspects of hepatic polarity, Huh7 are an unpolarized model of hepatocytes. Both lost NTCP expression, but overexpression of NTCP without restoring cellular polarity restores their susceptibility to infection. On the other hand, PHHs, differentiated HepaRG, and HLCs [65] display partial polarization and a high level of NTCP expression in vitro. While they are more mature models of polarized hepatocytes, they notoriously are less susceptible to HBV/HDV infection than depolarized cell lines overexpressing NTCP. This raises the question that cell polarity may actually restrict physical access of the viruses to NTCP. Early observations suggest that disruption of tight junctions by EGTA improved HBV infection of polarized dHepaRG [10], suggesting that in 2D cell culture, the viral receptor may be shielded by the tight junctions. However, in HLCs, such treatment did not improve infection by HDV [53]. Importantly, HLCs express NTCP at a higher level than PHHs in in vitro culture [51]. Moreover, NTCP is present on the surface of the HLCs exposed to the culture medium [53]. Recent advances in polarizing HLCs, by growing them on Transwell [66] or forming organoids [60], may constitute interesting new models to study how cell functional polarity affects HBV and HDV attachment, entry, and infection.

### 3.2. Heterogeneity, Differentiation, and Stability of Cell Populations

Notoriously, the different cell culture models to study HBV/HDV infection vary in terms of cellular maturity and population homogeneity. Hepatoma cell lines (Huh7 and HepG2) are particularly homogeneous but display an immature hepatic phenotype. On the other hand, mature models such as HLCs and dHepaRG form heterogeneous populations; HLCs exhibit various levels of hepatic maturation. Moreover, one single-cell RNAseq study showed the presence of non-hepatic cells within the culture of differentiated HLCs [67]. dHepaRG forms a dual cell population of hepatocyte and cholangiocyte-like cells [30], but how homogeneous the two populations are is not fully known. In this context, scRNAseq analysis of an HBV- or HDV-infected cell population may allow understanding how cellular heterogeneity drives or restricts infection.

As aforementioned, stem cell-derived HLCs and HepaRG undergoing hepatic differentiation acquire susceptibility to HBV and HDV infection as they start expressing NTCP. Interestingly, hepatoma cell lines stably overexpressing NTCP are readily susceptible to infection and seem to be influenced by in vitro treatment improving permissivity to HDV/HBV infection. For example, Huh7 pre-cultured with DMSO and HepG2 grown in iPSC-hepatocyte maintenance medium (HMM) [68] supports improved HBV infection.

The phenotypic stability of the different cell models is an important factor to consider when designing experiments. Hepatoma cell lines are very stable and support long-term culture and experiments. While PHHs rapidly dedifferentiate in vitro, dHepaRG and HLCs can be maintained for a few weeks in a monolayer, but lack of cell division ultimately limits their ability to support long-term infection. New developments in long-term maintenance of mature cell culture models, from micro-patterned PHHs [69] to coculture with mesenchymal cells and formation of liver organoids [60] seem to improve stability and unlock the way to long-term infection. However, the next challenge will be to make these complicated models accessible to most researchers.

### 3.3. Cell Proliferation and How Relevant It Is to Study HDV/HBV In Vitro

The HBV/HDV cell culture models vary greatly in terms of proliferative capacities. PHHs in vitro are quiescent; HLCs and dHepaRG become permissive to HBV/HDV after hepatic specification, a stage characterized with quiescence induction and start of hepatic maturation. On the other hand, non-differentiated HepaRG, HepaRG-NTCP, Huh7, and HepG2 are actively dividing. Overall, cell proliferation promotes viral spread by increasing the number of infected cells [70]. For example, HDV spreads through proliferation in immunodeficient hepatoma cell lines [70,71]. However, using immunocompetent HepaRG-NTCP, Zhang et al. showed that HDV replication is inhibited by the innate immune response of the host cell specifically during cell division [70]. Altogether, it remains to be assessed how relevant cell division-mediated spreading in vitro and innate immune clearance are.

### 3.4. Innate Immunity in Response to Infection

PHHs have been the Gold Standard for the study of virus-induced innate immunity. On the other hand, hepatoma cell lines are deficient in pathways related to innate immune sensing. Huh7 do not respond to viral infection, and HepG2 only partially. As an alternative to PHHs, HepaRG and HLCs represent innate immune-competent in vitro models that can be used to decipher the cellular response to viral infection.

Interestingly, there are discrepancies in terms of innate immune activation by HBV in these different cell models. HBV is usually believed to avoid detection by the cellular PRRs, thus does not trigger an innate immune response, as seen in the liver of chimpanzees and chronically infected patients [72], in primary hepatocytes [16], and in SC-derived HLCs [57]. On the other hand, innate triggering has been observed after baculovirus-driven transfection of the HBV genome in HepaRG [73,74]. However, later approaches based on bona fide HBV infection showed not triggering or inhibiting of the innate sensing in different cell culture models, including the differentiated HepaRG [36,57]. Interestingly, HBV replication is sensitive to a triggered innate immune response, as shown after in vitro super infection with HCV [75,76] and HDV [77] leading to inhibition of HBV replication by the HCV or HDV driven innate immunity.

All major immunocompetent models (PHHs, HepaRG, HLCs) trigger an innate immune response in response to HDV infection. Studies have shown that HDV is sensed by the PRR MDA5 [37], with help from LGP2 [78]. However, they are cytoplasmic sensors, while the main PAMP associated with HDV is believed to be intranuclear, where genome replication takes place. What component(s) of the virus triggers the innate immune response thus remains to be identified. The HDV-triggered innate immune response does not inhibit HDV replication in quiescent cell culture models [53]. However, in dividing HepaRG-NTCP cell lines, IFN treatment and innate immune activation seems to be able to restrict and sometimes clear the virus [70]. During cell division, the nucleus structure is disassembled, making the virus exposed to antiviral sensing and/or effectors in the cytoplasm. However, relevance of this observation in regard to mature quiescent models like PHHs, dHepaRG, and HLCs has to be investigated. HDV infection also seems to be sensitive to IFN at a very early stage of its viral cycle: Pre-treating cells with IFNα partially restricts HDV infection in PHHs [79], HepaRG-NTCP [37], and HLCs [53]; what mechanism is responsible for this restriction is unknown. Could it be similar to the one observed in dividing HepaRG cells, when the viral genome, normally sheltered in the nucleus, is released in the cytoplasm? Could cytoplasmic restriction mechanisms target the HDV RNP during its transit from uncoating to the nucleus? Much remains to be deciphered to truly understand how the host cell machinery may be able to efficiently control the infection.

Interestingly, the HDV-triggered innate immune response restricts the replication of its helper virus HBV in dHepaRG, consistent with clinical data that HDV super-infection partially diminishes HBV replication [77]. Such work emphasizes the importance of developing efficient in vitro co- or super-infection models in order to be able to analyze in detail the viruses’ crosstalk.

### 3.5. In Vitro Model of Efficient HBV/HDV Co-Infection

To understand the pathology and cellular interactions associated with chronic delta Hepatitis, it is important to reproduce accurately the HBV/HDV co-infection in vitro. However, this is difficult to achieve [80]. In hepatoma cell lines, HBV and HDV display different infection efficiency; HBV infects preferentially HepG2 while HDV replicates better in Huh7. Recent works showed that the HepG2 hepatoma cell line can support stable replication of both viruses simultaneously (HepG2BD) [81], however a hepatoma cell line-based model of co-infection relying on bona fide viral entry and infection remains to be established. While mature models like PHHs, dHepaRG, and HLCs seems to support a high level of infection by HBV, they support limited infection by HDV. In this context, a favored approach has been to first infect these cells with HBV, allowing establishment of a robust HBV replication after several days, and then to super-infect the HBV-infected cells with HDV. Such an approach has been performed in PHHs [16,82] and dHepaRG [77,82]. Preliminary results suggest that HLCs can also support co-infection HBV/HDV [55]. Improvements in cell culture conditions allowing long-term maintenance of mature hepatocytes, and better understanding of restriction mechanisms of HDV and HBV replication in the different cell culture models will probably make HDV/HBV in vitro co-infection more efficient and easier to achieve by the scientific community in the next years.

### 3.6. Co-Culture with Immune Cells

Actors of the immune system are believed to contribute to liver damages during chronic Hepatitis Delta [83]. Studying the interplay between immune cells and infected hepatocytes in vitro is thus of critical importance to understand the immunopathogenesis of Hepatitis Delta. However, reproducing such co-culture in vitro is complicated. HBV-infected 3D liver culture consisting of hepatocytes and Kupffer cells has been developed to study the interplay between the viral host cells and resident macrophages [84,85]. Interactions between infected hepatoma cell lines and peripheral immune NK [86] or T-cells [87,88] showed that mechanisms of immune control of HBV and HDV infection can be studied in vitro. Development of such complex co-culture models may help us understand mechanisms of clearance of hepatocytes with persistent forms of HBV infection (integrated genome and/or cccDNA) that has been resistant to treatments so far.

### 3.7. Genetic Background of the Host Cell and Personalized In Vitro Model

Observations that PHHs from different donors exhibit variable infection efficiency suggests that the genetic background of the donor may affect viral replication. Similar observations have been made using donor-derived iPSC-derived HLCs [60]. Contrary to cell line-based work that is limited to one genetic background, using patient-derived iPSC and their differentiation into various cell types, from macrophage to hepatocyte-like cells, may hold the key to develop individualized in vitro models of infection. However, how the natural history or the pathogenesis of the disease can be reproduced in vitro remains to be proven.

## 4. Conclusions

Relevant models to study viral infection before has led to the development of new therapeutic options that dramatically changed the rules for clinical practice and significantly benefited the patients. The field of HCV is such an example [19]. The goal to cure HBV and HDV infection in patients is still to be reached. Particularly, specific challenges like HDV maintenance in the cells even in presence of an IFN-based antiviral response and the establishment of extremely stable HBV cccDNA and integrated genome have to be accurately reproduced in vitro to have a chance to resolve them. The development of efficient and biologically relevant cell culture models leads us that way.

## Figures and Tables

**Figure 1 viruses-16-00716-f001:**
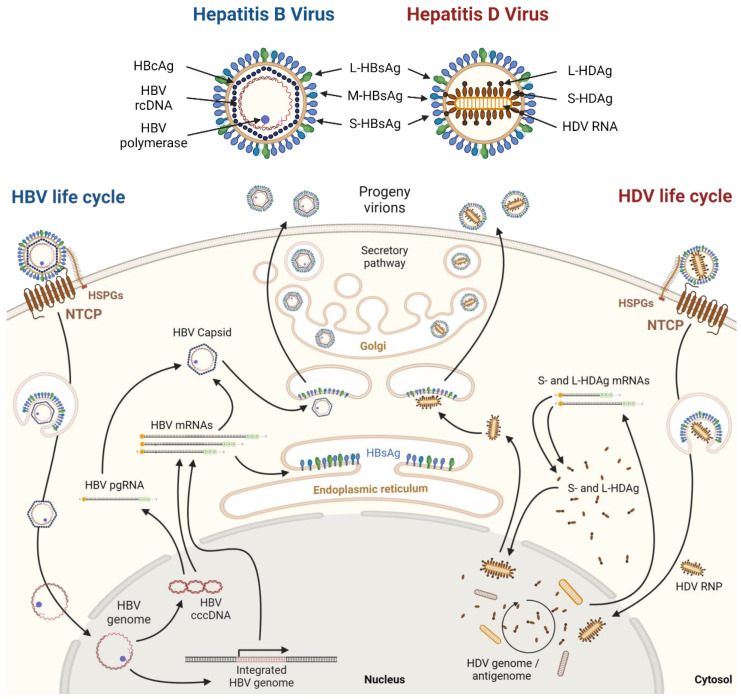
Simplified depiction of HBV and HDV viral particles and life cycles in hepatocytes. Created with Biorender.com.

**Figure 2 viruses-16-00716-f002:**
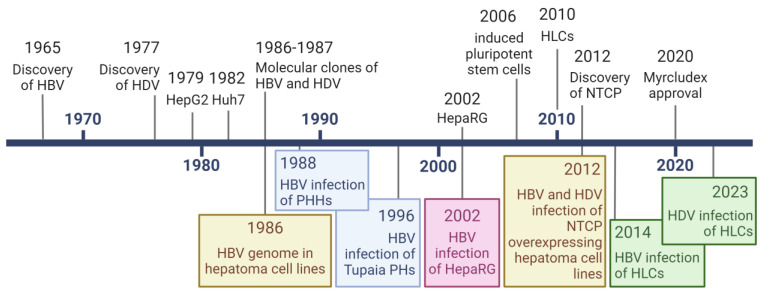
Timeline of HBV, HDV, and hepatic cell culture models breakthroughs. Created with Biorender.com.

**Table 1 viruses-16-00716-t001:** Summary of characteristics of the various HBV and HDV cell culture models.

Model	PHHs	Huh7/HepG2	HepaRG	dHepaRG	NTCP-Over Expressing Huh7/HepG2	HLCs
Viral entry	++	−	−	++	+++	++
HBV replication	+++	++/+++	++	++	+++	++
HDV replication	+++	+++/++	++	++	+++	+
HBV Production	+	++	+	+	++	++
HDV production	+	+++	+	+	+++	+
Advantages	Gold standard	Easy to handle	Easy to handle	Mature model	Easy to handle	Mature model
Limitations	Limited availability	Immature model	Immature model	Heterogeneous differentiation	Immature model	Difficult differentiation

(−) Inadequate, (+) poorly efficient, (++) efficient, (+++) highly efficient model.

**Table 2 viruses-16-00716-t002:** Characteristics of the different cell culture models related to points discussed below.

Model	PHHs	Huh7/HepG2	HepaRG	dHepaRG	HLCs
Stability	−	++	++	++	+
Maturity	++	−	−	+	+
Polarization	++	−/+	−	++	++
NTCP expression	++	−	−	+	++
Innate immunity	++	−/+	++	++	++
Proliferation	−	++	++	−	−

(−) limited, (+) partial, (++) good reproduction.

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
