# Peer review of "Cell Culture Models for Hepatitis B and D Viruses Infection: Old Challenges, New Developments and Future Strategies"

_viruses, 2024, doi:10.3390/v16050716_

Round 1

Reviewer 1 Report

Comments and Suggestions for Authors

In this review, Arnaud Carpentier gave a chronological overview of all in vitro model developed to study HBV and HDV infection and replication. Dr Carpentier emphasizes on the pros and cons of each model in regards of their efficiency, reproducibility, availability, innate immune response activity and how close they mimic human hepatocytes (maturity). The authors also briefly pinpoint to specific questions that remain to be answered in the field.

Overall, the review is very well done, easy to follow with the chronological display of the cell culture models that have been developed over the years. The limitations of each model system are clearly indicated.  Obviously, the ideal model system is still lacking, and scientists will be poised to use imperfect systems for some time.

I don’t have any major point but only few minor observations.

Minor points

Figure 1: Some of the arrows in the upper section of Fig 1 are difficult to follow.

The description of the engineered in vitro model (such as: HepNB2.7, Huh7-END, Huh7-2C8D) are not well described.

Section 3.5:  The new HepG2BD model recently reported by Blanchet et al. (HepG2BD: A Novel and Versatile Cell Line with Inducible HDV Replication and Constitutive HBV Expression. DOI:10.3390/v16040532) should be inserted in this section.

Author Response

I would like to thank reviewer 1 for his/her work reviewing the present review. Please find below a point-by-point answer to the comments raised. The manuscript has been corrected, and edits are written in red for easier reading.

Figure 1: Some of the arrows in the upper section of Fig 1 are difficult to follow.

I now removed the line fading on the arrows. I hope it makes it easier to read and follow.

 The description of the engineered in vitro model (such as: HepNB2.7, Huh7-END, Huh7-2C8D) are not well described.

I apologize for that. A typo made it difficult to understand the sentence. The paragraph has be reworked. I hope this is good enough.

Section 3.5:  The new HepG2BD model recently reported by Blanchet et al. (HepG2BD: A Novel and Versatile Cell Line with Inducible HDV Replication and Constitutive HBV Expression. DOI:10.3390/v16040532) should be inserted in this section.

I added this work to the section 3.5, as it describes that stable replication of both viruses is possible simultaneously in HepG2 cells. Thank you for making me aware of these new devlopments.

Reviewer 2 Report

Comments and Suggestions for Authors

The author provides a useful overview of cell culture systems used to study infections by HBV and HDV viruses that will be valuable to those in these research areas. There were a number of deficiencies, mostly minor but not unimportant, which are elaborated below:

Throughout this review, there are too many instances where appropriate references are not provided – some entire paragraphs contain no citations.

Line 43 – HBV assembly also involves the viral polymerase, which is associated with the RNA from which it was translated

Line 53 – The statement regarding the size of HDV is not clear. The HDV particle is not the smallest in size; rather, HDV has the smallest virus genome.

Lines 56 – 61. There are multiple problems with this summary of HDV replication. 

1.        As written, the author gives the impression that HDV is “deficient” because it does not encode an RNA-dependent RNA polymerase. All viruses depend on numerous host functions, so the reliance of HDV on the host for polymerase activity is not a deficiency at all. The only “deficiency” is the lack of an envelope protein, as the authors note  later.

2.        The genome is not the template for HDAg synthesis; the template for HDAg is the antigenome-sense mRNA. 

3.        The HDAg mRNA is not post-translationally modified. The modification – RNA editing at the amber/W site by ADAR1 – occurs on the antigenome during replication, not on the mRNA.

4.        There is not a single reference to the primary literature in this paragraph.

Line 71 – This description of the statistics on the outcome of HBV infection, should include the fact that it is remarkably age-dependent.

Line 72 – 73. The percent of HBV carriers positive for HDV superinfection varies considerably by region and there is a lot of discussion that current estimates might be too low because of insufficient testing.

Figure 2 – HDV was discovered in 1977, not 1972.

Lines 155 –157. References 26 and 27 describe the initiation of HDV replication by transfection of a plasmid containing 3 head-to-tail copies of an HDV cDNA clone, not a “full-length RNA”.

References 56 and 88 appear to be identical; neither reference is to a publication

Author Response

I would like to thanks Reviewer 2 for his/her careful work reviewing our review and allowing me to improve it. I would like first to explain why some sections, particularly the introduction, are showing no citations. I focused the review on the description of in vitro models, and thus only briefly brushed over the virology of both viruses, just mentioning what was needed to understand the following review. In this context, I made the choice to cite as an opening four reviews that provides a much more complete overview of the viruses’ biology (references 1 to 4). In order to not overload the reference sections, and limit the size of the manuscript, I did not mention every work quickly overlooked in the introduction. I hope this is acceptable.

Here is a point-by-point answer to the raised questions and comments:

Line 43 – HBV assembly also involves the viral polymerase, which is associated with the RNA from which it was translated

The paragraph has been rewritten to mention the presence of the HB polymerase in the capsid assembly.

Line 53 – The statement regarding the size of HDV is not clear. The HDV particle is not the smallest in size; rather, HDV has the smallest virus genome.

Sorry for this mistake, I now write that HDV is the “smallest RNA virus” to infect human.

Lines 56 – 61. There are multiple problems with this summary of HDV replication. 

I now realize that trying to keep the introduction concised led to potential confusion. I now rewrote some sections of it.

  1. As written, the author gives the impression that HDV is “deficient” because it does not encode an RNA-dependent RNA polymerase. All viruses depend on numerous host functions, so the reliance of HDV on the host for polymerase activity is not a deficiency at all. The only “deficiency” is the lack of an envelope protein, as the authors note  later.

I totally agree with reviewer 1. I now moved the sentence to later in the text, to not link deficiency and lack of RdRP.

  1. The genome is not the template for HDAg synthesis; the template for HDAg is the antigenome-sense mRNA. 

Thank you for pointing this error out. I now added mention to the antigenome of HDV and how important it is for both viral genome replication, and production of the L-HDAg. The antigenome has also been added to the figure 1.

  1. The HDAg mRNA is not post-translationally modified. The modification – RNA editing at the amber/W site by ADAR1 – occurs on the antigenome during replication, not on the mRNA.

Thank you for this comment, text has been amended to properly reflect this fascinating aspect of the HDV replication cycle.

  1. There is not a single reference to the primary literature in this paragraph.

As mentioned earlier, this is an author choice. The review focuses on model. In this context, I chose to limit the introduction to a quick overlook of virological concepts needed to understand the model sections. I only mention at the very beginning four recent reviews specifically focusing on life cycle of viruses, where the reader is invited to find more detailed description and original references. I hope the editorial committee and reviewers understand this choice.

Line 71 – This description of the statistics on the outcome of HBV infection, should include the fact that it is remarkably age-dependent.

The text has been modified to reflect this characteristic.

Line 72 – 73. The percent of HBV carriers positive for HDV superinfection varies considerably by region and there is a lot of discussion that current estimates might be too low because of insufficient testing.

Thank you for this comment: Under-testing of HDV is a key problem in many part of the world, and indeed influences the relevance of current epidemiological numbers. This point has been added to the manuscript.

Figure 2 – HDV was discovered in 1977, not 1972.

Sorry for this typo, the figure has been fixed.

Lines 155 –157. References 26 and 27 describe the initiation of HDV replication by transfection of a plasmid containing 3 head-to-tail copies of an HDV cDNA clone, not a “full-length RNA”.

Text has been corrected according to reviewer’s 2 comment.  

References 56 and 88 appear to be identical; neither reference is to a publication.

Sorry for a bug I should have detected. Mendeley did not properly generated reference from this special issue of JHep related to the EASL meeting where work by these authors was presented. While the work is not published yet, I considered it was worth mentioning it in this review. I now referenced the BioRXiv article related to this work. I hope this is OK.